# Serum Reactive Antibodies against the N-Methyl-D-Aspartate Receptor NR2 Subunit—Could They Act as Potential Biomarkers?

**DOI:** 10.3390/ijms242216170

**Published:** 2023-11-10

**Authors:** Maria S. Hadjiagapiou, George Krashias, Christina Christodoulou, Marios Pantzaris, Anastasia Lambrianides

**Affiliations:** 1Department of Neuroimmunology, The Cyprus Institute of Neurology and Genetics, Nicosia 2371, Cyprus; mariah@cing.ac.cy (M.S.H.); pantzari@cing.ac.cy (M.P.); 2Department of Molecular Virology, The Cyprus Institute of Neurology and Genetics, Nicosia 2371, Cyprus; georgek@cing.ac.cy (G.K.); cchristo@cing.ac.cy (C.C.)

**Keywords:** multiple sclerosis, N-methyl-D-aspartate (NMDA) receptor, antibodies, anti-coagulant serine proteases, excitotoxicity

## Abstract

Synaptic dysfunction and disrupted communication between neuronal and glial cells play an essential role in the underlying mechanisms of multiple sclerosis (MS). Earlier studies have revealed the importance of glutamate receptors, particularly the N-methyl-D-aspartate (NMDA) receptor, in excitotoxicity, leading to abnormal synaptic transmission and damage of neurons. Our study aimed to determine whether antibodies to the NR2 subunit of NMDAR are detected in MS patients and evaluate the correlation between antibody presence and clinical outcome. Furthermore, our focus extended to examine a possible link between NR2 reactivity and anti-coagulant antibody levels as pro-inflammatory molecules associated with MS. A cross-sectional study was carried out, including 95 patients with MS and 61 age- and gender-matched healthy controls (HCs). The enzyme-linked immunosorbent assay was used to detect anti-NR2 antibodies in serum samples of participants along with IgG antibodies against factor (F)VIIa, thrombin, prothrombin, FXa, and plasmin. According to our results, significantly elevated levels of anti-NR2 antibodies were detected in MS patients compared to HCs (*p* < 0.05), and this holds true when we compared the Relapsing-Remitting MS course with HCs (*p* < 0.05). A monotonically increasing correlation was found between NR2 seropositivity and advanced disability (r_s_ = 0.30; *p* < 0.01), anti-NR2 antibodies and disease worsening (r_s_ = 0.24; *p* < 0.05), as well as between antibody activity against NR2 and thrombin (r_s_ = 0.33; *p* < 0.01). The presence of anti-NR2 antibodies in MS patients was less associated with anti-plasmin IgG antibodies [OR:0.96 (95%CI: 0.92–0.99); *p* < 0.05]; however, such an association was not demonstrated when analyzing only RRMS patients. In view of our findings, NR2-reactive antibodies may play, paving the way for further research into their potential as biomarkers and therapeutic targets in MS.

## 1. Introduction

Multiple sclerosis (MS) is a chronic multifactorial disease of the central nervous system (CNS), characterized by inflammation, white matter demyelination, and neurodegeneration in the affected brain or spinal cord regions [1,2]. As of yet, there is no thorough knowledge of MS etiopathogenesis and the primary etiology of the disease is still elusive.

It has been shown that autoreactive T cells, macrophages, and microglial cells have a profound impact on the disease’s onset as they are key drivers of the inflammatory milieu in the CNS, causing myelin sheath destruction [3]. Recently, mechanisms that may precede neuroinflammation and are independent of immune responses have come to light. Notably, an excessive amount of glutamate in the CNS can have a detrimental effect, as it causes a sustained stimulation of the ionotropic glutamate N-methyl-D-aspartate receptor (NMDAR) on neurons and glial cells that leads to a cascade of abnormal signal transmission with excitotoxic and lethal effects [4]. Normally, the NMDAR is activated for a brief period, allowing extracellular cations, especially Ca^2+^, to influx into cells. This initiates signal transmission pathways, including synaptic plasticity, kinase and phosphatase activation, as well as cellular growth, survival or apoptosis, which are then regulated negatively by extracellular Mg^2+^ that inhibits the ion permeability of NMDAR [5,6]. Overactivation or sustained stimulation of the NMDAR leads to a dramatically increased influx of Ca^2+^ that further enhances the release of intracellularly stored Ca^2+^ and contributes to mitochondrial dysfunction, production of reactive oxygen species (ROS), and nitrogen species (RNS), oxidative stress, and iron deposition into the CNS [5,7]. NMDARs are expressed in both neurons and glial cells in the CNS; however, the regulation process for NMDARs differs significantly in each cell type. In neurons, the Mg^2+^-mediated blocking of the receptor is strictly regulated, whereas glial cells, and especially oligodendrocytes, are less sensitive, revealing that oligodendrocytes are dramatically affected in a glutamate overload followed by NMDAR sustained activation [8]. Similarly, ependymal cells demonstrate a high sensitivity to overactivation of NMDAR, which can affect the blood-brain barrier (BBB) integrity. Treatment with NMDAR antagonists like dizocilpine maleate (MK-801) can reduce neurotoxicity in the CNS and inhibit abnormal BBB permeability [5].

Current scientific literature also supports the presence of antibodies that can target the NMDAR subunits, i.e., GR1N1 (NR1) and GR1N2 (NR2), and serve as potential biomarkers for significantly increased cognitive impairment in Parkinson’s disease [9], neuronal injury in amyotrophic lateral sclerosis (ALS) and ischaemic stroke [10,11], or neuropsychiatric symptoms in anti-NMDAR encephalitis [12], as well as in autoimmune diseases like systemic lupus erythematosus (SLE), associated with early cell death in the presence of high NR2-reactive antibody concentration [10,13]. Particularly, in SLE, a five-peptide consensus sequence: Asp/Glu-Trp-Asp/Glu-Tyr-Ser/Glu (DWEYS) was identified in a DNA mimotope, and the same pattern was also found in the amino acid 283–287 region of the extracellular ligand-binding domain (LBD) of the NR2 subunit. Antibodies against the DNA mimotope can cross-react with the particular NR2 domain, impairing receptor function, especially in neuropsychiatric SLE (NPSLE) cases, and leading to cognitive deficits [14,15].

Interestingly, antibodies against the NR1 subunit can reduce NMDAR surface density in a titer-dependent manner without affecting other synaptic channels [16]. This holds true when assessing the NMDAR synaptic localization, as antibodies against the NR1 subunit interfere with the synaptic signal transmission mediated by Ca^2+^ influx. On the other hand, AMPA and other synaptic channels do not exhibit lack of synaptic localization or non-active conformation [16].

Moreover, NR2-reactive antibodies have been associated with ischaemic stroke in hypertensive patients rather than haemorrhagic stroke and serve as potential biomarkers of cardiovascular diseases (CVD) [11]. Thus, low levels of anti-NR2 antibodies have been defined as an indicator of improved disease phenotype, whereas increased levels of anti-NR2 antibodies indicate a worse prognosis [15]. Importantly, CVD, i.e., hypertension, myocardial infarction, and atherosclerosis, are significant contributors to ischaemic strokes in MS, especially during the early stages [17,18]. Given that a higher anti-NR2 antibody titer is a marker of cerebral ischemia in individuals at high CVD risk [19,20], NR2-reactive antibodies should be investigated in MS pathogenic mechanisms. The current study attempted to identify seropositivity to NR2 in patients diagnosed with MS and assess the implication of the particular reactive antibodies in disease clinical outcomes.

Previously, we have also shown that a significant proportion of MS patients (43%) was positive for antibodies against coagulant serine proteases, functioning as procoagulant-pro-inflammatory mediators and contributing further to the abnormal coagulation-inflammation interplay in MS [21,22]. Notably, a reciprocal relationship between coagulation and inflammation is observed following the activation of microglia and the infiltration of immune cells in the CNS [23]. Coagulation-inflammation activation is characterized by overexpression of the Tissue Factor (TF) transmembrane receptor in response to vessel injury, inflammatory cytokines, and the expression of damaged-associated molecular patterns, forming the TF-FVIIa complex with the circulating factor VII [24,25]. This complex activates FX, followed by the generation of thrombin from its zymogen prothrombin [24], which cleaves fibrinogen to fibrin [26]. Fibrin clots can also be formed by activating the FXI-intrinsic coagulation pathway through FXII activity [27]. In pathological conditions, thrombin triggers pro-inflammatory signaling pathways and leukocyte migration into the CNS through the activation of thrombin-activated protease-activated receptors (PARs) [28]. In addition, fibrin clots are bound to integrin MAC-1 (CD11b/CD18) receptors on microglial cells, activating microglia and promoting pro-inflammatory cytokine production [29]. Therefore, our goal was to determine whether activity against the NR2 subunit can be correlated with activity against the serine proteases that play a critical role in the coagulation-inflammation interaction in MS [22].

## 2. Results

### 2.1. Clinical and Nonclinical Data of Study Participants

One hundred and fifty-six individuals were enrolled, providing data on both demographic and clinical features. As shown in Table 1, the ratio of female to male individuals diagnosed with MS was 1.7, similar to the MS patient ratio of 1.6 female to male reported in Cyprus [30]. The age range of MS patients varies from 21 to 80 years, showing no significant difference with the age range of controls (*p* = 0.06); similarly, the gender between the two study groups was not significantly different (*p* = 0.31). The average duration of the disease was 16.23 ± 9.4 years, while different stratifications of patients were revealed when evaluating disease disability and severity. Therefore, 47.4% of patients had mild MS disability (EDSS: 0–3.0), 35.8% of MS participants showed moderate disability (EDSS: 3.5–5.5), and 16.8% had severe disability status scale (EDSS: 6.0–9.5). In addition, the disease progression indicated by the MSSS score showed a median index of 3.4 (Interquartile Range: 2.4–5.5), with 10.5% of patients stratified into aggressive disease progression (MSSS: 7–10). Thus, different types of medication, including immunosuppressive and immunomodulatory drugs, were prescribed according to the medical history of MS participants, whereas a significant percentage of patients (27.4%) did not receive any prescribed medication for MS disease. Table 1 also shows the clinical and nonclinical features of the RRMS course solely since there were analyses conducted specifically in this MS type. RRMS patients and HCs were matched in age (*p* = 0.53) and gender (*p* = 0.38). Fifty-six percent of patients had mild MS disability (EDSS: 0–3.0), 34.7% of MS participants showed moderate disability (EDSS: 3.5–5.5), and 9.3% had severe disability status scale (EDSS: 6.0–9.5). In addition, the disease progression indicated by the MSSS score showed a median index of 3.05 (Interquartile Range: 2.1–4.3), with 6.7% of patients stratified into aggressive disease progression (MSSS: 7–10).

### 2.2. Qualitative Analysis of Antibody Binding Activity against NR2 Subunit

The antibody binding activity against the NR2 subunit of the NMDA receptor was examined in an indicative number of 95 MS patients, and the results were compared with those obtained from 61 healthy controls.

As Figure 1 demonstrates, MS patients exhibited a higher mean ratio index of a sample’s O.D. to the O.D. of the negative control [mean: 1.90 (standard error of the mean; SEM: 0.28)] than healthy participants [mean: 1.70 (SEM: 0.11)]. This difference significantly distinguished the two study groups, with an AUC prediction value of 0.60 (95% CI: 0.50–0.69; *p* < 0.05).

However, when performing multiple comparisons between the courses of MS disease and HCs, i.e., RRMS-HCs; RRMS-progressive types of MS (SPMS and PPMS); and progressive types of MS-HCs using the post hoc Dunn’s test, no significant difference was found, as shown in Table 2. At this point, it should be noted that due to the limited number of SPMS and PPMS patients analyzed in the current study and the fact that both disease courses are more progressive and less inflammatory than the RRMS course, individuals with SPMS or PPMS were encompassed in the category of progressive types of MS.

Results are expressed as the ratio of sample O.D./negative control O.D. (manufacturer’s instructions). SEM: Standard error of the mean; RRMS: Relapsing-Remitting MS; SPMS: Secondary Progressive; PPMS: Primary Progressive; HCs: Healthy controls; ns: not significant. The Kruskal–Wallis test was performed for the analysis and Dunn’s test for the multiple comparisons.

Nevertheless, based on the Mann–Whitney U test, RRMS demonstrated a considerably higher mean activity to NR2 [mean: 1.80 (SEM: 0.26)] in comparison to healthy controls [mean: 1.70 (SEM: 0.11); *p* < 0.05], when evaluated as two independent groups (Figure 2). This outcome possesses an AUC discrimination value of 0.60 (95% CI: 0.50–0.70; *p* < 0.05), and it is imperative to be considered for further analysis.

Overall, 11 out of 75 RRMS patients (14.7%) showed seropositive status against the NR2 subunit, as opposed to 1 of 19 patients with progressive MS phenotype (5.2%) and 10 seropositive HCs (16.4%) (Figure 3).

### 2.3. Correlation between Anti-NR2 Antibody Levels and Clinical/Laboratory Outcomes of MS Patients

Previously, we have reported the presence of IgG antibodies against serine proteases associated with the coagulation cascade in MS patients [21]. The activity against FVIIa, thrombin, prothrombin, FXa, and plasmin was examined in our line of work to further analyze whether such antibody presence is correlated with anti-NR2 antibody detection. The results of anti-serine protease antibodies were expressed as the percentage of each sample’s binding activity compared to the binding activity of a reference sample. Table 3 shows the mean indices of binding activity levels for the five antibodies in MS patients as well as in different MS subgroups.

Further analysis was conducted using the Spearman correlation coefficient to determine whether anti-NR2 antibodies were correlated with antibodies against coagulant molecules in MS patients, to assess any possible correlation between various clinical and nonclinical features and to find differences in outcome rankings (Table 4). A monotonically increasing correlation was found between anti-NR2 and anti-FXa activity levels (r_s_ = 0.22, *p* < 0.05) (Figure 4A) and between anti-NR2 and anti-thrombin IgG (r_s_ = 0.33, *p* < 0.01) (Figure 4B). On the other hand, a negative correlation was observed between anti-NR2 and anti-FVIIa IgG levels (r_s_ = −0.22, *p* < 0.05) (Figure 4C). In addition, a positive correlation was observed between anti-NR2 activity levels and EDSS (r_s_ = 0.30, *p* < 0.01), as well as between anti-NR2 and MSSS (r_s_ = 0.24, *p* < 0.05).

Logistic regression analysis was also performed to evaluate the association between the seropositivity to the NR2 subunit and the clinical and laboratory findings of MS patients, and the odds ratio (OR) was obtained (Table 5). According to the OR, the presence of antibodies against plasmin in MS patients is less associated with the NR2 activity [OR: 0.96 (95% CI: 0.92–0.99); *p* < 0.05]; however, such an association in RRMS patients was not observed when analyzing the anti-NR2 antibody levels with anti-plasmin IgG [OR: 0.99 (95% CI: 0.93–1.00)] (Table 6).

### 2.4. Analysis between Anti-NR2 Antibody Levels and Receiving Medications for MS Treaatment

MS patients receiving medication and those not receiving medication were compared to HCs in terms of binding activity against the NR2 subunit, considering MS treatment. (Table 7). Using the post hoc Dunn’s test, there was no significant difference observed between MS patients receiving medication and HCs, MS patients without medication and HCs, or MS patients receiving medication and patients without medication.

However, when analyzing particularly the NR2-reactive antibody levels between patients who received medication and HCs by the Mann–Whitney test, a significantly higher mean activity for NR2 was revealed in patients [mean: 1.78 (SEM: 0.29)] compared to healthy controls [mean: 1.68 (SEM: 0.11); *p* < 0.05].

Upon conducting a more in-depth analysis, it was demonstrated that MS patients taking immunosuppressive medication had higher levels of antibodies (mean: 1.77; SEM: 0.44) compared to healthy controls (mean: 1.68; SEM: 0.11; *p* < 0.05). This was not observed in MS patients taking immunomodulatory medication and compared with controls (Table 8). As Figure 5 shows, NR2 reactivity levels varied significantly between the patients with immunosuppressive drugs and controls (Figure 5). On the other hand, logistic regression analysis did not reveal an association between antibody presence and medication received in MS group [OR: 0.69 (95% CI: 0.32–1.2)].

## 3. Discussion

MS is a chronic complex disease of the CNS that engages not only with an inflammatory phenotype and demyelination, but also with the dysfunction of synapses in gray matter and the disruption of neuronal-glial communication [31,32]. Overstimulation of glutamate receptors in MS, especially the ionotropic NMDARs, can provoke synaptic perturbation and lead to altered network dynamics in the CNS. Consequently, impulse transmission is impaired, providing cognitive dysfunction and disability [2].

The current study attempted to shed some light on the link between antibodies directed against the NR2 subunit of the NMDAR and MS disease, as considerable evidence suggests the implication of NR2-reactive antibodies in inflammatory demyelinating diseases [33]. Therefore, we aimed to compare the levels of anti-NR2 antibodies in MS patients with different ranks of disease progression, with those found in HCs. According to our findings, significantly elevated anti-NR2 antibody levels were shown in MS patients compared to HCs; the same also holds true when comparing the activity levels between the RRMS course and HCs solely. With respect to the scientific literature available, only a few studies have been conducted on NR2-reactive antibodies, highlighting that such molecules may serve as mediators of non-inflammatory but detrimental effects in autoimmune and demyelinated diseases [34,35]. Of note, isolated antibodies against the NR2 subunit, derived from patients with SLE, were injected into the brain tissue of mouse models and led to an increased rate of neuronal apoptosis without previously developing inflammation [36]. Importantly, anti-NR2 antibodies contribute to brain injury following BBB disruption. Abnormally increasing BBB permeability leads to the import of such antibodies from the periphery to the CNS, primarily affecting the transmission of axon potential to nerves and disturbing cell interactions [37]. Conversely, the intact BBB prevents the import of such molecules and their subsequent effects on CNS tissue [37]. 

Current scientific literature provides evidence that NR2-reactive antibody presence is an indicator of ischaemic stroke or transient ischaemic attack (TIA) [11], while increased serum anti-NR2 titer was associated with white matter hyperintensities in the brain of patients suffering from arterial hypertension [38]. Along with the aforementioned findings, activity against the NR2 subunit was also reported in patients with cerebral small vessel disease (SVD), resulting in cognitive impairments and presence of lesions in white matter. In particular, Dobrynina et al. revealed that an increased titer of anti-NR2 antibodies is most likely to be detected as a consequence of hypoxia-mediated BBB disruption and serve as biomarker of SVD, especially during the early stages of the disease [39].

At this point, it is worth mentioning that ischaemic stroke, cardiovascular diseases, arterial hypertension, SVD, and hyperintensities in white matter are common findings in MS [40]. Since earlier studies have documented the implication of anti-NR2 in such diseases and pathological conditions, the role of anti-NR2 should also be considered for MS pathology. Moreover, NR2-reactive antibodies act in a dose-dependent manner, affecting the synaptic localization of NMDAR, synaptic transmission, and plasticity, which play an essential role in the pathophysiology of MS [16]. Such molecules stimulate, among others, the dysfunction of neuronal networks, neuronal injury, and cell death, contributing to disease exacerbation [9,13], therefore NR2-reactive antibodies should also be evaluated for further research in MS. To our knowledge, this is the first study aiming to assess the clinical and laboratory features of MS patients with respect to NR2-reactive antibodies. Previous work by Ramberger and colleagues [34] did not show significant findings when assessing anti-NR2 antibodies in demyelinated diseases, including MS; therefore, further investigation is required to evaluate in depth the role of such antibodies in the underlying pathological mechanisms of MS.

In addition, the present study revealed a correlation of antibody binding activity to NR2 with advanced disability and disease progression in addition to the relationship of anti-NR2 antibody levels with thrombin reactivity. Paired with our previous work, documenting an increased likelihood of disease severity with the presence of thrombin-reactive IgG antibodies [21], this provides thorough knowledge of antibody implication in disease progression from the perspective of anti-NR2 antibody identification.

On the other hand, anti-FVIIa IgG antibodies showed a negative correlation with anti-NR2 antibody presence, an observation that warrants further investigation. In accordance with our earlier work [21], anti-FVIIa IgG antibody levels were negatively correlated with IgG antibodies against coagulant serine proteases that were characterized as pro-inflammatory molecules and potential mediators of the coagulation-inflammation interplay in MS [22]. Even though it is unclear how FVIIa-reactive IgG antibodies play a role in underlying pathological mechanisms [35,36], it may be worthwhile to consider the inverse relationship between FVIIa- and NR2-reactive antibodies that may result in an improved MS phenotype or decreased exacerbation.

It is noteworthy that activity to the NR2 subunit was less associated with anti-plasmin IgG in MS patients; however, such an association was not significantly shown when the analysis was conducted solely in the RRMS group of patients. Of note, the presence of anti-plasmin IgG was associated not only with advanced disability but also with the elderly age of MS participants in our previous work [21]. Comparably, we speculate that NR2-reactivity has an increased likelihood of being recognized during the early stages of the disease, especially as a biomarker in the RRMS course, while plasmin-reactivity is more relevant to progressive states of MS.

Plasmin activation and NMDAR degradation are both catalyzed by the tissue plasminogen activator (t-PA) [37], which contributes to fibrinolysis [41] but can also mediate diverse effects in the CNS. In particular, expression of t-PA in neuronal cells regulates synaptic plasticity and cell survival; however, it may induce neuroinflammation through microglial cell activation [42]. Moreover, t-PA may also provide either NMDA-mediated neuroprotective effects by suppressing Ca^2+^ influx into the neurons or neurotoxic effects by modulating the overstimulation of the NMDAR, leading to neuronal damage [43,44]. Under pathological conditions, the released NR1 and NR2 subunits following the NMDAR cleavage by t-PA constitute a triggering factor for the activation of the immune system and the subsequent production of antibodies against the aforementioned subunits [45]. To the best of our knowledge, this is the first study that engages the relationship between anti-NR2 and anti-plasmin antibodies; therefore, further research is needed to validate our findings and to analyze in depth the role of these molecules along with t-PA function in neuroinflammatory and demyelinating diseases.

In this context, it is important to note that the current study is not without limitations, requiring further investigation to draw robust conclusions. The first limitation was the small number of participants, which reduced the statistical power of the study and made it difficult to determine the accuracy of the test. As a result, the anti-NR2 levels discriminated poorly between MS patients and controls (AUC = 0.60), affecting the reliability of the results. Due to the heterogeneity of our cohort tested (duration of the disease, treatment received, MS courses) and the fact that MS patients were blindly selected, the RRMS patients were overrepresented compared with the SPMS and PPMS in our cohort tested. This was more likely to decrease the power of the survey and show a weak ability of cohort discrimination. Future studies are needed to validate our results by increasing the number of participants and drawing robust conclusions by analyzing particular parameters that would be consistent with more precise and reproducible data. Nevertheless, even a model with an AUC of 0.6 may still prove useful in revealing the role of such molecules in autoimmune neuroinflammatory diseases. Furthermore, it is imperative though, to investigate the role of such antibodies and characterize the signaling pathways in which anti-NR2 antibodies are involved to further understand their implication in MS. In addition, it should also be examined whether the antibody production is consistently detected in patients throughout the years by performing retrospective analysis and to assess the persistent seropositivity to the NR2 subunit with respect to MS progression. Moreover, we should also refer that another limitation of our study was the upset of the correlation of antibody presence with clinical key aspects like the association of vascular risk factors that could influence the results obtained and could also be a confounding factor when determining IgG antibodies against serine proteases associated with the coagulation cascade in MS patients. Similarly, the evaluation of the relapse phase of the disease, which is the active inflammatory disease phase, should also be considered based on the antibody concentration levels. Furthermore, in future research the anti-NR2 antibody levels will be assessed based on neuroimaging aspects, especially the activity of active lesions with gadolinium uptake. 

## 4. Materials and Methods

### 4.1. Study Participants

Serum samples were obtained from 156 donors for the implementation of the protocols below; one individual suffered from clinically isolated syndrome (CIS); 75 individuals were diagnosed with Relapsing-Remitting MS (RRMS); 16 individuals were diagnosed with Secondary Progressive MS (SPMS); three were diagnosed with Primary Progressive MS (PPMS); and 61 were healthy controls (HCs).

All participants diagnosed with MS were followed up at the Neuroimmunology Department of the Cyprus Institute of Neurology and Genetics between May 2016 and January 2018, satisfying all of McDonald’s revised criteria for inclusion [46]. To be enrolled in this study, all individuals were required to meet the following criteria: age above 18 years old; a sufficiently defined disease course [Relapsing-Remitting MS, Secondary Progressive MS, Primary Progressive MS]; and clearly defined clinical measurements [Expanded disability status scale (EDSS); MS severity score (MSSS)]. The researchers were blinded to MS courses. Unwillingness to provide informed consent, a history of alcoholic or drug abuse, and pregnancy were all considered exclusion criteria.

The participants were thoroughly informed about the study objectives and filled out a consent form approved by the Cyprus National Bioethics Committee (ΕΕΒΚ/ΕΠ/2016/51).

### 4.2. Serum Collection

Serum was isolated within three hours of fresh blood collection and stored at −20 °C until further processing. Previously, blood samples were collected in vacutainers without any additive, and were allowed to clot followed by centrifugation at 1500× *g* for 10 min at 25 °C.

### 4.3. Determination of Human Glutamate [NMDA] Receptor Subunit Epsilon-2 (NR2) Antibody

An immunoassay technique was employed for the detection of antibodies against the epsilon-2 (NR2) subunit of the NMDA receptor using a commercial enzyme-linked immunosorbent assay (ELISA) (Cusabio, Wuhan, China).

As per protocol, the kit was run for 30 min at 37 °C with a positive control, a negative control, and the samples of interest, followed by washes to remove any unbound reagent. Half-hour incubation with Horseradish Peroxidase-conjugated anti-human immunoglobulin was then carried out at 37 °C. Following a 30-min incubation with the appropriate substrate at 37 °C, the signal developed proportionally to the presence of bound anti-NR2 antibodies on the pre-coated plate.

The results were determined qualitatively using the manufacturer’s guidelines. The antibody activity was expressed as the ratio of a sample’s O.D. to the O.D. of the negative control. When the ratio was equal to or greater than 2.1, the sample was considered positive; and when the index was less than 2.1, the sample was considered negative.

### 4.4. Qualitative Determination of Activity to Coagulant Serine Proteases

In-house, indirect enzyme-linked immunosorbent assay (ELISA) protocols were performed to determine qualitative antibody binding activity directed against five serine proteases associated with the coagulation pathway. As previously described [21], activity against factor (F) VIIa, thrombin (Thr), prothrombin (PT), FXa, and plasmin was measured, and the results of the antibody binding activity of samples were determined as the percentage of the average net absorbance of the sample tested divided by the absorbance of a reference sample. Each sample’s net absorbance was calculated by subtracting the absorbance detected on the control side, which had only diluent added, from the absorbance detected on the test side, which had an antigen diluted in a specific diluent.

#### 4.4.1. Activity to FVIIa

MaxiSorp plates (ThermoFisher, Scientific, Waltham, MA, USA) were used for the detection of IgG antibody directed against FVIIa. Plates were coated with 50 μL of 1.5 μg/mL human FVIIa diluted in phosphate-buffered saline (PBS) at the test side of the plate, while PBS alone was added at the control side and incubated overnight at 4 °C. Plates were then blocked with 200 μL of PBS/2% bovine serum albumin (BSA) at 37 °C for 90 min and washed three times with PBS. Serum samples were diluted at 1:25 in PBS/1% BSA and incubated at 25 °C for 60 min. Conjugated anti-human IgG diluted to 1:1000 in PBS/1% BSA was incubated in a total of 50 μL per well. After one hour of incubation at RT and three washes in PBS, tetramethylbenzidine chromogenic solution (TMB) was added for 30 min followed by 1 M HCl stop solution, and the absorbance value was read at 450 nm [47].

#### 4.4.2. Activity to Thrombin

For the analysis of IgG antibodies directed against thrombin, an ELISA assay using Costar plates (ThermoFisher, Scientific, Waltham, MA, USA) was performed. Plates were coated with 50 μL of 10 μg/mL human alpha-thrombin in Tris-buffered saline (TBS; 0.05 M Tris, 0.15 M NaCl; pH 7.6) and incubated overnight at 4 °C. After three washes in TBS, wells were blocked with TBS/0.3% gelatin at RT for 90 min. Previously, gelatin was dissolved in the blocking solution by high-temperature heating. Assay samples were incubated for 90 min at RT in TBS/0.1% gelatin and anti-human IgG conjugated with HRP-detected bound IgG. After one hour of incubation, TMB substrate was added followed by 1 M HCl stop solution and absorbance reading at 450 nm [47].

#### 4.4.3. Activity to Prothrombin

MaxiSorp plates were coated with 10 μg/mL of human prothrombin in TBS. After the overnight incubation at 4 °C, wells were blocked with 1% BSA in TBS at RT for two hours. Serum samples diluted 1:25 in TBS/1% BSA were on wells and incubated for 90 min. Labeled anti-human IgG was used for bound IgG detection followed by substrate addition and read of absorbance at 450 nm [48].

#### 4.4.4. Activity to FXa

A concentration of 5 μg/mL FXa in TBS coated the test side of 96-well Costar plates, while diluent was added alone on the control side, and the plates were incubated overnight at 4 °C. To avoid non-specific binding, the wells were blocked with TBS containing 0.3% gelatin and incubated at RT for 90 min. Serum samples were diluted 1:25 in TBS containing 0.1% gelatin and incubated for 90 min. HRP-conjugated anti-human IgG was then incubated for one hour, and a colorimetric reaction with TMB was followed. After adding the stop solution, the absorbance was measured at 450 nm [49].

#### 4.4.5. Activity to Plasmin

For the analysis of activity against plasmin serine protease, a concentration of 5 μg/mL plasmin in PBS coated the half side of Costar plates, while on the other half only the diluent was added. After the overnight incubation at 4 °C and washing in PBS, 0.25% gelatin in PBS was used to block the uncoated sides of the wells and incubated at RT for 60 min. Samples diluted in PBS/0.1% gelatin were dispensed into the wells followed by incubation at RT for 90 min and detection of bound IgG using HRP-conjugated anti-human IgG in PBS/0.1% gelatin. TMB substrate was added, and the colorimetric reaction was measured at 450 nm [50].

### 4.5. Statistical Analysis

The GraphPad Prism V.5.00 for Windows software program (La Jolla, CA, USA) was employed for the statistical analysis. The normality test was conducted by the D’Agostino–Pearson test, and the analysis for age- and gender-matching between patients and controls was performed by the non-parametric Mann–Whitney U test and Fisher’s exact test, respectively. The non-parametric Mann–Whitney U test was applied also to compare differences in antibody distribution between the two study groups. Kruskal–Wallis one-way ANOVA followed by Dunn’s post hoc test for multiple comparisons was used to assess antibody presence in more than two groups of participants. The accuracy of the test was evaluated using the area under a Receiver Operating Characteristic (ROC) curve (AUC). The values range from 0.5, which corresponds to no discrimination, to 1.0, which corresponds to perfect discrimination between the two groups tested. A logistic regression analysis was conducted to assess the antibody activity regarding baseline characteristics. A correlation matrix was performed using the Spearman correlation coefficient to analyze the association between antibody activity levels. The values range from −1, which corresponds to the perfect negative correlation to +1, which is the perfect positive correlation.

## 5. Conclusions

In conclusion, our study sought to shed light on the presence of antibodies directed against the NR2 subunit of the NMDA glutamate receptor in MS. Antibody titers were significantly higher in MS patients than in healthy controls, which were associated with disease severity and disability in patients. The activity of NR2 was also correlated with antibody activity levels against coagulant components, which contribute to disease progression, according to our previous studies. In view of these findings, further studies are needed to shed light on anti-NR2 antibodies’ impact on MS pathophysiology, providing new insights in the perturbation of synaptic mechanisms and signal transmission, as well as ablation of NMDAR function.

## Figures and Tables

**Figure 1 ijms-24-16170-f001:**
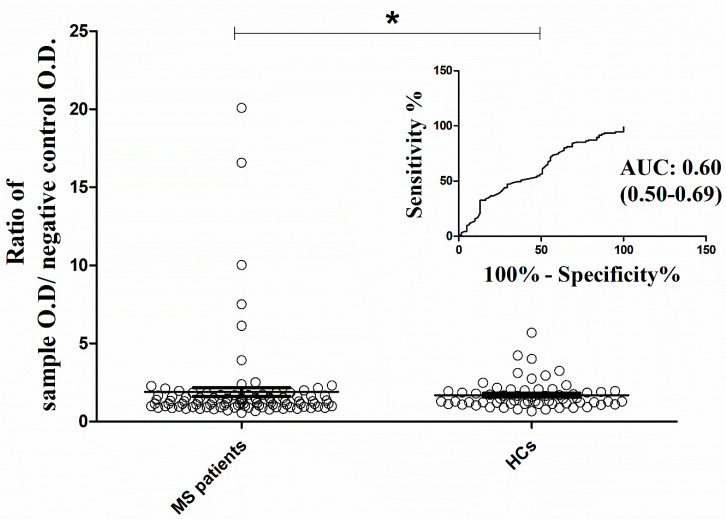
Distribution of antibodies against the NR2 subunit of the NMDAR between MS patients and healthy controls. The antibody activity was expressed as the ratio of a sample’s O.D. to the O.D. of the negative control. Mann-Whitney test applied for the analysis (* *p* < 0.05). AUC was measured for accuracy of the test MS: Multiple Sclerosis; NMDAR: N-Methyl-D-aspartate receptor; AUC: area under the curve.

**Figure 2 ijms-24-16170-f002:**
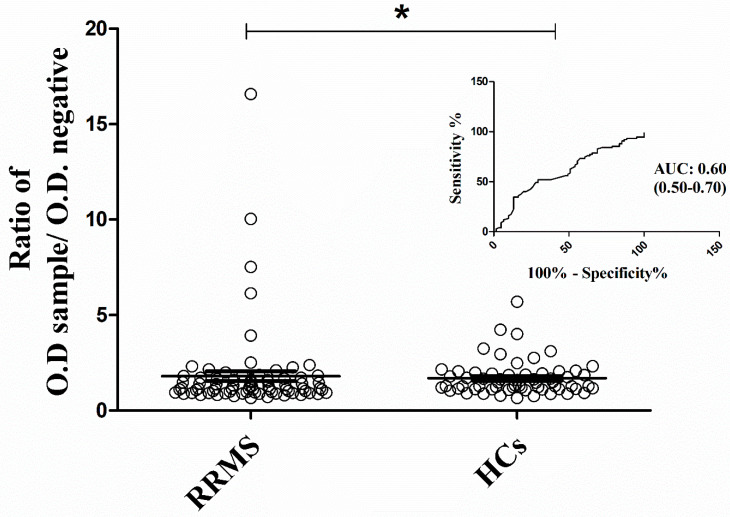
Immunoglobulin activity against the NR2 subunit of the NMDAR between RRMS patients and healthy controls. The antibody activity was expressed as the ratio of a sample’s O.D. to the O.D. of the negative control. Mann-Whitney test applied for the analysis (* *p* < 0.05). AUC was measured for accuracy of the test. RRMS: Relapsing-Remitting Multiple Sclerosis; NMDAR: N-Methyl-D-aspartate receptor; AUC: area under the curve.

**Figure 3 ijms-24-16170-f003:**
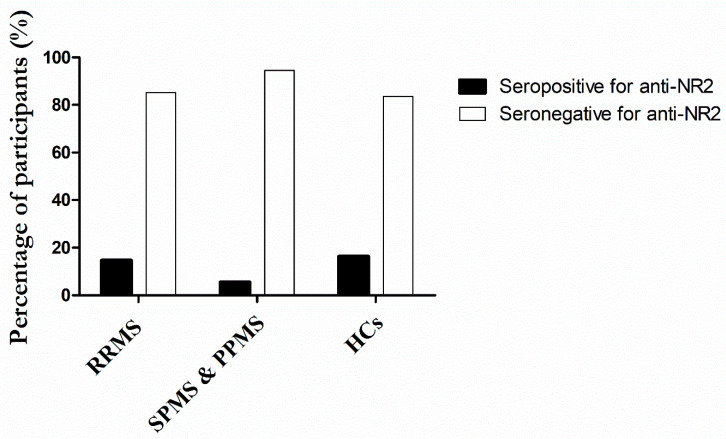
Comparative prevalence (%) of IgG detection in MS patients and HCs. Results are expressed as the percentage of participants tested positive (bars shown in black) or negative (bars shown in white) for the presence of anti-NR2 antibodies. Seropositivity was determined as follows: when the ratio was equal to or greater than 2.1, the sample was considered positive; and when the index was less than 2.1, the sample was considered negative.

**Figure 4 ijms-24-16170-f004:**
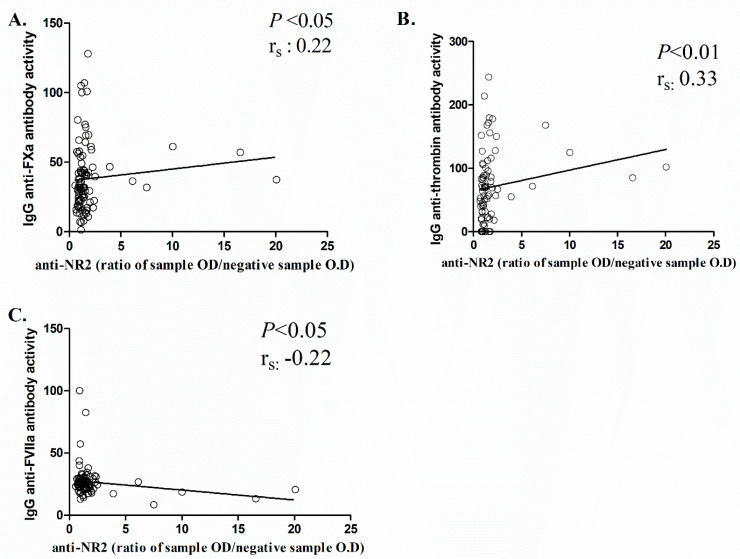
Correlation between anti-NR2 antibodies and antibody activity against coagulant components. Positive correlation was observed between anti-NR2 and anti-FXa activity levels (**A**) and between anti-NR2 and anti-thrombin IgG (**B**). Negative correlation was demonstrated between anti-NR2 and anti-FVIIa IgG levels (**C**). The Spearman correlation coefficient was applied for the analysis. Values are expressed as the ratio of a sample’s O.D. to the O.D. of the negative control. Data with statistical significance are shown.

**Figure 5 ijms-24-16170-f005:**
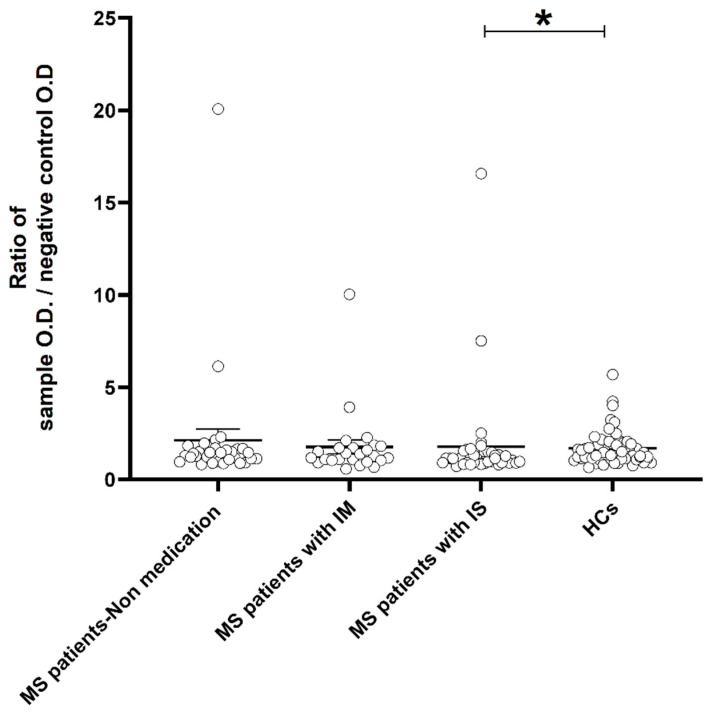
Binding activity against the NR2 subunit of the NMDAR based on the receiving medication for MS treatment. A significant difference was observed when assessing the ratio of NR2-reactive antibody values from patients receiving immunosuppressive medication with those obtained from HCs. Bars represent the mean ± SEM. The Kruskal-Wallis test was performed for the analysis and Dunn’s test for the multiple comparisons (* *p* < 0.05). MS: Multiple Sclerosis; IM: Immunomodulatory medication; IS: Immunosuppressive medication; HCs: healthy controls.

**Table 1 ijms-24-16170-t001:** Demographic and clinical profile of participants.

Features	MS Patients (*n* = 95)	HCs (*n* = 61)	*p* Value	RRMS (*n =* 75)	*p* Value
Gender					
Female/Male	60/35	33/28	0.31	47/28	0.38
Age in years					
Mean ± SD	48 ± 14	43.5 ± 12	0.06	45.4 ± 13	
Min–Max	21–80	23–66		21–80	0.53
Disease course (CIS/RRMS/SPMS/PPMS)					
	1/75/16/3	N/A			
Disease Duration (years)				15.16 ± 9.6	
Mean ± SD	16.23 ± 9.4	N/A		15 (9–21)	
Median (interquartile range)	17 (11–22)				
EDSS					
Median (interquartile range)	3.5 (2.25–4.80)	N/A		3.0 (2.0–4.0)	
Mild: 0–3.0 [*n* (%)]	45 (47.4)			42 (56.0)	
Moderate: 3.5–5.5 [*n* (%)]	34 (35.8)			26 (34.7)	
Severe: 6.0–9.5 [*n* (%)]	16 (16.8)			7 (9.3)	
MSSS					
Median (interquartile range)	3.4 (2.4–5.5)	N/A		3.05 (2.1–4.3)	
Benign MS: 1–2 [*n* (%)]	17 (17.9)			15 (20.0)	
Severe MS: 7–10 [*n* (%)]	10 (10.5)			5 (6.7)	
Medication [*n* (%)]		N/A			
Interferon beta-1a or -1b	20 (21.0)			20 (26.7)	
Natalizumab	12 (12.7)			12 (16.0)	
Fingolimod	17 (17.9)			16 (21.3)	
Other *	20 (21.0)			10 (13.3)	
None	26 (27.4)			17 (22.7)	

MS: Multiple Sclerosis; HCs: Healthy controls; CIS: Clinically Isolated Syndrome; RRMS: Relapsing-Remitting MS; SPMS: Secondary Progressive MS; PPMS: Primary Progressive MS; SD: Standard Deviation; EDSS: Expanded Disability Status Scale; MSSS: Multiple Sclerosis Severity Score; N/A: Not Applicable. * Other type of treatment: Azathioprine, Dimethyl fumarate, Glatiramer acetate, Methotrexate, Mycophenolate, Rituximab, Teriflunomide.

**Table 2 ijms-24-16170-t002:** Binding activity against NR2 subunit of NMDA receptor with respect to disease courses.

Antibodies	Mean Index (SEM)			Kruskal–Wallis Test *p* Value	Dunn’s Multiple Comparisons Test		
	RRMS	Progressive Types of MS (SPMS and PPMS)	HCs		RRMS-HCs	Progressive Types of MS-HCs	RRMS-Progressive Types of MS
					*p* Value		
anti-NR2	1.80 (0.26)	2.26 (0.99)	1.69 (0.11)	=0.10	ns	ns	ns

**Table 3 ijms-24-16170-t003:** Antibody activity to serine proteases associated with the coagulation cascade in MS patients.

	Anti-FVIIa	Anti-FXa	Anti-PT	Anti-Thrombin	Anti-Plasmin
	Mean Index (SEM)				
MS patients	26.70 (1.20)	38.18 (2.50)	9.70 (1.60)	71.17 (6.30)	39.54 (2.40)
RRMS patients	26.46 (1.27)	38.28 (2.76)	9.48 (1.71)	72.15 (7.10)	38.22 (2.56)
SPMS and PPMS patients	28.08 (3.28)	38.19 (6.11)	9.04 (1.71)	66.70 (15.22)	43.62 (6.10)

FVIIa: Activated factor VII; FXa: activated factor X; PT: prothrombin; MS: Multiple Sclerosis; RRMS: Relapsing-Remitting MS; SPMS: Secondary Progressive; PPMS: Primary Progressive; SEM: Standard Error of the Mean.

**Table 4 ijms-24-16170-t004:** Correlation analysis between the presence of antibodies against NR2 and various laboratory and clinical outcomes.

	Laboratory Finding					Demographic and Clinical Findings			
	Anti-FVIIa	Anti-FXa	Anti-PT	Anti-Thrombin	Anti-Plasmin	Age	Disease Duration	EDSS	MSSS
	Spearman Correlation Coefficient (95% Confident Interval) *p* Value								
Anti-NR2	−0.22 (−0.41–0.01) *p* = 0.03	0.22 (0.02–0.41) *p* = 0.03	−0.06 (−0.27–0.14) *p* = 0.54	0.33 (0.11–0.52) *p* = 0.003	0 (−0.20–0.21) *p* = 0.95	0.20 (−0.01–0.39) *p* = 0.06	0.12 (−0.09–0.32) *p* = 0.24	0.30 (0.10–0.48) *p* = 0.0028	0.24 (0.03–0.43) *p* = 0.02

FVIIa: Activated factor VII; FXa: activated factor X; PT: prothrombin; EDSS: Expanded Disability Status Scale; MSSS: Multiple Sclerosis Severity Score; CI: Confidence Interval. All the correlation analyses were performed using the Spearman correlation coefficient.

**Table 5 ijms-24-16170-t005:** Logistic regression analysis between NR2 antibody binding activity and demographic or clinical features of MS patients.

	Gender (ref. M)	Age	EDSS	MSSS	Anti-FVIIa	Anti-FXa	Anti-PT	Anti-Thrombin	Anti-Plasmin	RRMS	Progressive Types (SPMS and PPMS)
Anti-NR2	**Odds Ratio (95% CI)**										
	0.79 (0.23–2.88)	1.00 (0.96–1.05)	0.95 (0.66–1.34)	0.93 (0.66–1.26)	0.90 (0.81–1.0)	1.0 (0.98–1.03)	0.98 (0.92–1.02)	1.00 (0.99–1.02)	0.96 (0.92–0.99) *p* = 0.03	3.27 (0.57–61.59)	0.33 (0.02–1.86)

**Table 6 ijms-24-16170-t006:** Logistic regression analysis between NR2 antibody binding activity and demographic or clinical features of RRMS patients.

	Gender (ref. M)	Age	EDSS	MSSS	Anti-FVIIa	Anti-FXa	Anti-PT	Anti-Thrombin	Anti-Plasmin
Anti-NR2	**Odds Ratio (95% CI)**								
	0.67 (0.18–2.56)	1.00 (0.96–1.06)	0.95 (0.60–1.41)	0.87 (0.58–1.21)	0.91 (0.80–1.0)	1.01 (0.98–1.03)	0.98 (0.92–1.03)	1.0 (0.99–1.02)	0.97 (0.93–1.00)

M: Male; EDSS: Expanded Disability Status Scale; MSSS: Multiple Sclerosis Severity Score; CI: Confidence Interval; FVIIa: Activated factor VII; FXa: activated factor X; PT: prothrombin; RRMS: Relapsing-Remitting Multiple Sclerosis; SPMS: Secondary Progressive; PPMS: Primary Progressive.

**Table 7 ijms-24-16170-t007:** Binding activity against NR2 subunit of NMDA receptor with respect to MS treatment.

Antibodies	Mean Index (SEM)			Kruskal–Wallis Test *p* Value	Dunn’s Multiple Comparisons Test		
	Non-Medication	Medication	HCs		Non-Medication-HCs	Non-Medication-Medication	Medication-HCs
					*p* Value		
anti-NR2	2.12 (0.62)	1.78 (0.29)	1.68 (0.11)	=0.07	ns	ns	ns

SEM: Standard Error of Mean; HCs: healthy controls.

**Table 8 ijms-24-16170-t008:** Binding activity against NR2 subunit of NMDA receptor with respect to immunomodulatory and immunosuppressive treatment.

Antibodies	Mean Index (SEM)				Kruskal–Wallis Test *p* Value	Dunn’s Multiple Comparisons Test				
	Non-Medication	IM Medication	IS Medication	HCs		Non-Medication-IM	Non-Medication-IS	IM-IS Medication	IM-HCs	IS-HCs
						*p* Value				
anti-NR2	2.12 (0.62)	1.76 (0.37)	1.77 (0.44)	1.68 (0.11)	=0.06	ns	ns	ns	ns	0.04

IM: Immunomodulatory medication; IS: Immunosuppressive medication; SEM: Standard Error of Mean; HCs: healthy controls.

## Data Availability

The data that support the findings of this study are available from the corresponding author upon reasonable request.

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
