# Peer review of "Serum Reactive Antibodies against the N-Methyl-D-Aspartate Receptor NR2 Subunit—Could They Act as Potential Biomarkers?"

_ijms, 2023, doi:10.3390/ijms242216170_

Round 1
Reviewer 1 Report
Comments and Suggestions for Authors
This study aimed to to determine whether antibodies to the NR2 subunit of NMDAR are detected in MS patients and evaluate the correlation between antibody presence and clinical outcome. Below are my comments:
1. The authors should clearly distinguish the unfavorable effects of the stimulation or persistent activation of glutamate N-methyl-D-aspartate receptor (NMDAR) on neurons and glial cells receptor from its blocking by the presence of antibodies against the NMDAR subunits. These are two fundamental issues that are mutually exclusive and cannot be presented as arguments for the purpose of undertaking research and discussing the results. It is commonly assumed that NMDAR antibodies cause a selective decrease in NMDAR surface density (see doi: 10.1523/JNEUROSCI.0167-10.2010).
2. The structure of the article is surprising - the results and discussion sections precedes the material and methods section.
3. AUC discrimination value of 0.60 is relatively low and requires comments.
4. The presence of NR2-reactive antibodies in many diseases (Parkinson's disease, ALS, ischaemic stroke, SLE) is a factor that negatively influences their potential importance as a biomarker in MS.
5. The authors reported that ,, increased serum anti-NR2 titer was associated with white matter hyperintensities in the brain of patients suffering from arterial hypertension”. Hence my question: how many patients in their group had hypertension or other cerebrovascular diseease, because all of MS individuals had hyperintense lesions in CNS.
6. In the authors' opinion, should the detection of NR2 antibodies not lead to re-verification of the diagnosis of MS and consideration of, for example, SVD.
7. The conclusions drawn that anti-NR2 antibodies may serve as biomarkers for MS monitoring and prognosis as well as that it would be interesting to consider such antibodies and the NR2 antigen as therapeutic strategies in MS are too far-reaching at this stage.
Author Response
Reviewer 1:
Comment 1. The authors should clearly distinguish the unfavorable effects of the stimulation or persistent activation of glutamate N-methyl-D-aspartate receptor (NMDAR) on neurons and glial cells receptor from its blocking by the presence of antibodies against the NMDAR subunits. These are two fundamental issues that are mutually exclusive and cannot be presented as arguments for the purpose of undertaking research and discussing the results. It is commonly assumed that NMDAR antibodies cause a selective decrease in NMDAR surface density (see doi: 10.1523/JNEUROSCI.0167-10.2010).
We have now enriched the section of Introduction with the paragraphs below (lines 53-104):
Recently, mechanisms that may precede neuroinflammation and are independent of immune responses have come to light. Notably, an excessive amount of glutamate in the CNS can have a detrimental effect, as it causes a sustained stimulation of the ionotropic glutamate N-methyl-D-aspartate receptor (NMDAR) on neurons and glial cells that leads to a cascade of abnormal signal transmission with excitotoxic and lethal effects [4]. Normally, the NMDAR is activated for a brief period, allowing extracellular cations, especially Ca2+, to influx into cells. This initiates signal transmission pathways, including synaptic plasticity, kinase and phosphatase activation, as well as cellular growth, survival or apoptosis, which are then regulated negatively by extracellular Mg2+ that inhibit the ion permeability of NMDAR [5,6] Overactivation or sustained stimulation of the NMDAR leads to a dramatically increased influx of Ca2+ that further enhances the release of intracellularly stored Ca2+ and contributes to mitochondrial dysfunction, production of reactive oxygen species (ROS), and nitrogen species (RNS), oxidative stress, and iron deposition into the CNS [5,7]. NMDARs are expressed in both neurons and glial cells in the CNS; however, the regulation process for NMDARs differs significantly in each cell type. In neurons, the Mg2+ -mediated blocking of the receptor is strictly regulated, whereas glial cells, and especially oligodendrocytes, are less sensitive, revealing that oligodendrocytes are dramatically affected in a glutamate overload followed by NMDAR sustained activation [8]. Similarly, ependymal cells demonstrate a high sensitivity to overactivation of NMDAR, which can affect the blood-brain barrier (BBB) integrity. Treatment with NMDAR antagonists like dizocilpine maleate (MK-801) can reduce neurotoxicity in the CNS and inhibit abnormal BBB permeability [5].
Current scientific literature also supports the presence of antibodies against that can target the NMDAR subunits, i.e., GR1N1 (NR1) and GR1N2 (NR2), and serve as potential biomarkers for significantly increased cognitive impairment in Parkinson's disease [9] or neuronal injury in Amyotrophic lateral sclerosis (ALS), and ischaemic stroke [10,11], as well as in autoimmune diseases like systemic lupus erythematosus (SLE), associated with early cell death in the presence of high NR2-reactive antibody concentration [10,12]. Particularly, in SLE, a five-peptide consensus sequence: Asp/Glu-Trp-Asp/Glu-Tyr-Ser/Glu (DWEYS) was identified in a DNA mimotope, and the same pattern was also found in the amino acid 283-287 region of the extracellular ligand-binding domain (LBD) of the NR2 subunit. Antibodies against the DNA mimotope can cross-react with the particular NR2 domain, impairing receptor function, especially in neuropsychiatric SLE (NPSLE) cases, and leading to cognitive deficits [11,12].
Interestingly, antibodies against the NR1 subunit can reduce NMDAR surface density in a titer-dependent manner without affecting other synaptic channels [13]. This holds true when assessing the NMDAR synaptic localization, as antibodies against the NR1 subunit interfere with the synaptic signal transmission mediated by Ca2+ influx. On the other hand, AMPA and other synaptic channels do not exhibit lack of synaptic localization or non-active conformation [13].
Similarly, we added a paragraph in discussion to emphasize the antibody role: (lines 356-361): Moreover, NR2-reactive antibodies act in a dose-dependent manner, affecting the synaptic localization of NMDAR, synaptic transmission, and plasticity, which play an essential role in the pathophysiology of MS [15]. Such molecules stimulate, among others, the dysfunction of neuronal networks, neuronal injury, and cell death, contributing to disease exacerbation [9,12], therefore NR2-reactive antibodies should also be evaluated for further research in MS.
Comment 2. The structure of the article is surprising – the results and discussion sections precedes the material and methods section.
We followed the suggested IJMS manuscript template, which places the methodology section after the discussion. Please advise.
Comment 3. AUC discrimination value of 0.60 is relatively low and requires comments.
We have included some comments for the low value of AUC in the section of the Discussion, especially in the paragraph stating this limitation (lines 326-340). At this point, we would like also to point out that the participants for the present study were selected from the same group of participants, who were involved in our previous projects [e.g., doi:10.1016/j.msard.2022.103775]. Consequently, we had a limited pool of individuals that we were able to analyze.
(Lines: 409-423) In this context, it is important to note that the current study is not without limitations, requiring further investigation to draw robust conclusions. The first limitation was the small number of participants, which reduced the statistical power of the study and made it difficult to determine the accuracy of the test. As a result, the anti-NR2 levels discriminated poorly between MS patients and controls (AUC = 0.60), affecting the reliability of the results. Due to the heterogeneity of our cohort tested (duration of the disease, treatment received, MS courses) and the fact that MS patients were blindly selected, the RRMS patients were overrepresented compared with the SPMS and PPMS in our cohort tested. This was more likely to decrease the power of the survey and show a weak ability of cohort discrimination. Future studies are needed to validate our results by increasing the number of participants and drawing robust conclusions by analyzing particular parameters that would be consistent with more precise and reproducible data. To our knowledge, this is the first study that has demonstrated the presence of anti-NR2 antibodies in MS. In this regard, even a model with an AUC of 0.6 may still prove useful in revealing the role of such molecules in autoimmune neuroinflammatory diseases.
Comment 4. The presence of NR2-reactive antibodies in many diseases (Parkinson's disease, ALS, ischaemic stroke, SLE) is a factor that negatively influences their potential importance as a biomarker in MS.
As already mentioned in the manuscript, antibodies against the NR2 subunit of NMDAR have been detected in various neurological and autoimmune diseases with the role of biomarkers in the pathophysiology of these diseases
[doi:10.1176/appi.neuropsych.20220107, doi:10.1007/s00415-011-6232-5, doi:10.1016/j.nbd.2020.105161].
NR2-reactive antibodies should also be candidate markers for further research in MS since such antibodies act in a dose-dependent manner, affecting the synaptic localization of NMDAR, synaptic transmission, and plasticity, which play an essential role in the pathophysiology of MS and stimulate, among others, the dysfunction of neuronal networks, neuronal injury, and cell death, contributing to disease exacerbation.
A broader research analysis with a large sample size of RRMS, SPMS, and PPMS patients will also be useful to assess the frequency of NR2-reactive antibodies in each MS course and to identify whether they show a persistent activation that might contribute to the neurodegenerative and demyelinating phenotype of a particular MS course. This finding could also be significant in clinical routine from the perspective of personalized medicine and targeted therapeutic strategy, as each patient is characterized by specific clinical and laboratory features and constitutes a unique case in the study of MS heterogeneity.
It is also of great importance the overlapping cases of anti-NMDAR encephalitis and MS, in which anti-NR2 antibodies play a role, giving a significant effort to the diagnosis, especially when there are atypical symptoms, and the therapeutic approach that has to be followed in overlapping cases [doi: 10.3389/fimmu.2023.1088801].
Therefore, we suggest that anti-NR2 antibodies may serve as potential biomarkers in the study of MS, as they may play a role in the degeneration of neurons and injury of glial cells, and must be investigated in regards to the antibody titer concentration as well as the patient-specific clinical outcomes (e.g., disease course, disease duration, laboratory findings that may enhance the function of NR2-reactive antibodies, and treatment received).
In lines 356-361 we have added the paragraph below: Moreover, NR2-reactive antibodies act in a dose-dependent manner, affecting the synaptic localization of NMDAR, synaptic transmission, and plasticity, which play an essential role in the pathophysiology of MS (Hughes et al., 2010). Such molecules stimulate, among others, the dysfunction of neuronal networks, neuronal injury, and cell death, contributing to disease exacerbation (Gibson et al., 2023; Lauvsnes & Omdal, 2012), thereforeNR2-reactive antibodies should also be evaluated for further research in MS.
Comment 5. The authors reported that, increased serum anti-NR2 titer was associated with white matter hyperintensities in the brain of patients suffering from arterial hypertension”. Hence my question: how many patients in their group had hypertension or other cerebrovascular disease, because all of MS individuals had hyperintense lesions in CNS.
We reported some of the findings of the published paper: “Circulating autoantibodies against the NR2 peptide of the NMDA receptor are associated with subclinical brain damage in hypertensive patients with other pre-existing conditions for vascular risk” [doi: 10.1016/j.jns.2017.02.028]. The authors evaluated the levels of serum anti-NR2 antibodies in 47 patients with arterial hypertension to assess the antibody levels and whether they could act as predictors of white matter hyperintensities and severe brain MRI changes and evaluate their relationship with the occurrence of stroke and cognitive decline. We mentioned the findings of this study to emphasize a potent association of serum anti-NR2 antibodies with brain damage and stroke development in diseases with clinical features similar to MS (stroke, arterial hypertension, white matter hyperintensities), especially during the early stages of MS. Thus, we attempted to investigate the presence of reactive NR2 antibodies in MS. In future studies, we aim to further analyze the clinical data of the patients in response to increased NR2 titers. As it stands, we are unable to obtain this information from patients as this was not the initial scope of the study; an updated approval from the Cyprus bioethics committee will be required.
Comment 6. In the authors' opinion, should the detection of NR2 antibodies not lead to re-verification of the diagnosis of MS and consideration of, for example, SVD.
Antibodies against the NR2 subunit have been used in neurological or autoimmune diseases as biomarkers of synaptic dysfunction, cognitive impairments, and cell death, even though as predictors of brain damage and the development of clinical findings that could be associated with stroke onset. Nevertheless, and to our knowledge, physicians re-verify the diagnosis of each disease, using all clinical and laboratory findings to have reliable and valid results.
Similarly, we believe that anti-NR2 antibodies can be used as a significant tool for better monitoring and prognosis of MS. The detection of anti-NR does not exclude previously established long-term clinical and laboratory analyses that verify or re-verify MS onset. Instead, they can play a role as an additional tool for a better classification of patients with severe and non-severe clinical outcomes and an additional predictor of disease progression and exacerbation. Moreover, identifying antibodies against the NR2 subunit can aid in developing more precise therapeutic strategies based on the unique clinical and laboratory characteristics of each patient. This approach can target potential interactions between the antibodies and other molecules, which may impact the severity of the disease progression. Of course, all these are very early in current clinical practice and too far reaching at this stage, therefore further research is needed to study in depth the presence of such molecules and their impact in MS pathophysiology.
Comment 7. The conclusions drawn that anti-NR2 antibodies may serve as biomarkers for MS monitoring and prognosis as well as that it would be interesting to consider such antibodies and the NR2 antigen as therapeutic strategies in MS are too far-reaching at this stage.
We have altered the conclusions section as follows:
(lines: 565-568): In conclusion, our study sought to shed light on the presence of antibodies directed against the NR2 subunit of the NMDA glutamate receptor in MS. Antibody titers were significantly higher in MS patients than in healthy controls, which were associated with disease severity and disability in patients. The activity of NR2 was also correlated with antibody activity levels against coagulant components, which contribute to disease progression, according to our previous studies. In view of these findings, further studies are needed to shed light on anti-NR2 antibodies impact on MS pathophysiology, providing new insights in the perturbation of synaptic mechanisms and signal transmission, as well as ablation of NMDAR function.
Reviewer 2 Report
Comments and Suggestions for Authors
Although the topic is new and there are no previous studies with antibodies against the N-methyl-D-aspartate 2 receptor NR2 subunit in multiple sclerosis.
We have to consider that the work has sufficient limitations so that it cannot be accepted since the changes necessary for its correction represent a very important modification of the work, in the conception and execution. That would mean proposing the research again.
It needs to take into account that :
1) In the introduction there is no clear mention of the alteration in coagulation and its relationship to the pathogenesis of MS.
2) In material and methods, lines 321 and 323 appear data that should go in the results section
3) We understand that key aspects such as the presence of relapses that could influence the presence of antibodies reactive to NR2 (active inflammatory forms), nor neuroimaging aspects with the activity of active lesions with Gd uptake, have not been taken into account.
4) The treatment received can clearly influence the appearance of antibodies reactive to NR2. The authors could have classified the patients according to the treatment or into two groups, patients with treatment or without treatment, assuming that these are 27.4%, representing a high percentage of patients without treatment, fundamentally this occurs in the RRMS forms.
5) These NR2-reactive antibodies are related to cardiovascular risk factors such as hypertension, acute myocardial infarction or stroke. It should be analyzed whether patients with MS are associated with vascular risk factors that could influence the results obtained, and could also be a confounding factor when determining IgG antibodies against serine proteases associated with the coagulation cascade in MS patients.
6) The sample is composed of a number of patients with a small “n”, 94 patients with only 16 SPMS and 3 PPMS, the progressive forms (SPMS and PPMS) being those that are not associated with the presence of NR2-reactive antibodies. Precisely the progressive forms have a longer evolution time and greater disability measured by the EDSS. Measurements that are associated in this work with NR2-reactive antibodies.
7) Therefore, the conclusions of the work are a bit risky in terms of its biomarker potential. Furthermore, being a cross-sectional study, we cannot establish causality.
Author Response
Reviewer 2
Comment 1. In the introduction there is no clear mention of the alteration in coagulation and its relationship to the pathogenesis of MS.
The introduction has been amended and now mentions the alteration in coagulation and its relationship to the pathogenesis of MS as suggested by the reviewer.
(Lines: 119-132): Notably, a reciprocal relationship between coagulation and inflammation is observed following the activation of microglia and the infiltration of immune cells in the CNS [21] Coagulation-inflammation activation is characterized by overexpression of the Tissue Factor (TF) transmembrane receptor in response to vessel injury, inflammatory cytokines, and the expression of damaged-associated molecular patterns, forming the TF-FVIIa complex with the circulating factor VII [22,23]. This complex activates FX, followed by the generation of thrombin from its zymogen prothrombin [22] which cleaves fibrinogen to fibrin [24]. Fibrin clots can also be formed by activating the FXI-intrinsic coagulation pathway through FXII activity [25]. In pathological conditions, thrombin triggers pro-inflammatory signaling pathways and leukocyte migration into the CNS through the activation of thrombin-activated protease-activated receptors (PARs) [26]. In addition, fibrin clots are bound to integrin MAC-1 (CD11b/CD18) receptors on microglial cells, activating microglia and promoting pro-inflammatory cytokine production [27].
Comment 2. In material and methods, lines 321 and 323 appear data that should go in the results section.
We have removed the data in lines 321-323 and we added them in the section of the results (lines: 141-143) as proposed: The age range of MS patients varies from 21 to 80 years, showing no significant difference with the age range of controls (p=0.06); similarly, the gender between the two study groups was not significantly different (p=0.31).
Comment 3. We understand that key aspects such as the presence of relapses that could influence the presence of antibodies reactive to NR2 (active inflammatory forms), nor neuroimaging aspects with the activity of active lesions with Gd uptake, have not been taken into account.
Our principal aim was to detect the presence of antibodies against the NR2 subunit in MS patients and compare the results with those from individuals derived from the general population. The MS patients at the time of sampling were not in the relapse phase; however, it would be interesting to correlate the results with relapse cases. In addition, because we are a research laboratory, it wasn’t easy to ask for neuroimaging aspects at that particular time of sampling from the patients and physicians. However, since our results have provided insights and we aim to further analyze the presence of antibodies against the subunits of NMDAR, we have taken into account your advice as part of a wider effort to increase the broader range of clinical and non-clinical aspects that have to be considered in the study of antibodies against the NMDAR subunits.
For the current study, we have mentioned all the above as limitations and what we are planning to assess in future studies as follows: (lines: 367- 375): Moreover, we should also refer that another limitation of our study was the upset of the correlation of antibody presence with clinical key aspects like the association of vascular risk factors that could influence the results obtained, and could also be a confounding factor when determining IgG antibodies against serine proteases associated with the coagulation cascade in MS patients. Similarly, the evaluation of the relapse phase of the disease, which is the active inflammatory disease phase, should also be considered based on the antibody concentration levels. Furthermore, in future research the anti-NR2 antibody levels will be assess based on neuroimaging aspects, especially the activity of active lesions with gadolinium uptake.
Comment 4. The treatment received can clearly influence the appearance of antibodies reactive to NR2. The authors could have classified the patients according to the treatment or into two groups, patients with treatment or without treatment, assuming that these are 27.4%, representing a high percentage of patients without treatment, fundamentally this occurs in the RRMS forms.
We have analyzed the antibody levels based on the medication patients were receiving for MS treatment and we introduced another one subsection in the section of the Results, showing this analysis. Moreover, two Tables (Tables 7 and 8) and one Figure (Figure 5), demonstrating our results have now been included.
(Lines: 267-288) 2.4. Analysis between anti-NR2 antibody levels and receiving medications for MS treatment
MS patients receiving medication and those not receiving medication were compared to HCs in terms of binding activity against the NR-2 subunit, considering MS treatment. (Table 7). Using the post-hoc Dunn’s test, there was no significant difference observed between MS patients receiving medication and HCs, MS patients without medication and HCs, or MS patients receiving medication and patients without medication.
However, when analyzing particularly the NR2-reactive antibody levels between patients who received medication and HCs by the Mann-Whitney test, a significantly higher mean activity for NR2 was revealed in patients [mean: 1.78 (SEM: 0.29)] compared to healthy controls [mean: 1.68 (SEM: 0.11); p<0.05].
Upon conducting a more in-depth analysis, it was demonstrated that MS patients taking immunosuppressive medication had higher levels of antibodies (mean: 1.77; SEM: 0.44) compared to healthy controls (mean: 1.68; SEM: 0.11; p<0.05). This was not observed in MS patients taking immunomodulatory medication and compared with controls (Table 8). As Figure 5 shows, NR-2 reactivity levels varied significantly between the patients with immunosuppressive drugs and controls (Figure 5). On the other hand, logistic regression analysis did not reveal an association between antibody presence and medication received in MS group [OR: 0.69 (95% CI: 0.32-1.2)].
Comment 5. These NR2-reactive antibodies are related to cardiovascular risk factors such as hypertension, acute myocardial infarction or stroke. It should be analyzed whether patients with MS are associated with vascular risk factors that could influence the results obtained, and could also be a confounding factor when determining IgG antibodies against serine proteases associated with the coagulation cascade in MS patients.
As previously mentioned, we are a research laboratory part of the Clinical Department of the Cyprus Institute of Neurology and Genetics. Unfortunately, within the time frame given for the revisions of this manuscript, we were not allowed to access the medical records of the 95 patients who participated in this study, due to data protection rules. This also required an additional approval from the Bioethics Committee. Nevertheless, the reviewer’s valuable advice has been taken into consideration; in future studies, we will investigate the potential of a possible influence in antibody production with the clinical and laboratory features of patients and how these are also affecting the progression and exacerbation of the disease. In the current study, we mentioned as a limitation the upset of comorbidities and other risk factors associated with CVD with the aim to analyze them in further research.
(lines: 431-439): Moreover, we should also refer that another limitation of our study was the upset of the correlation of antibody presence with clinical key aspects like the association of vascular risk factors that could influence the results obtained, and could also be a confounding factor when determining IgG antibodies against serine proteases associated with the coagulation cascade in MS patients. Similarly, the evaluation of the relapse phase of the disease, which is the active inflammatory disease phase, should also be considered based on the antibody concentration levels. Furthermore, in future research the anti-NR2 antibody levels will be assess based on neuroimaging aspects, especially the activity of active lesions with gadolinium uptake.
Comment 6. The sample is composed of a number of patients with a small “n”, 94 patients with only 16 SPMS and 3 PPMS, the progressive forms (SPMS and PPMS) being those that are not associated with the presence of NR2-reactive antibodies. Precisely the progressive forms have a longer evolution time and greater disability measured by the EDSS. Measurements that are associated in this work with NR2-reactive antibodies.
The selection of MS patients for the current study was blinded; therefore, we could not address from the beginning the equal number of patients classified as RRMS, SPMS, and PPMS. After recruitment of patients, we identified that the majority of the patients stratified as RRMS. Our main goal was the identification of antibodies against NR2 in MS patients, which is why we did not insist specifically on an equal number; however, as we mentioned in the last paragraph of the discussion, this is a limitation that we will monitor in future studies. Based on our analysis, we found 11 out of 75 RRMS patients to be positive for the antibodies studied, and only one PPMS was characterized as seropositive.
Subsequently, when analyzing the EDSS and MSSS severity score, we found that there was a correlation between antibody presence and increased EDSS score or MSSS, showing that greater advanced disability (EDSS) or severe progression of the disease, given by the MSSS score (which is calculated taking into account the EDSS and the duration of the disease), are associated with increased antibody titer (Table 4).
Comment 7. Therefore, the conclusions of the work are a bit risky in terms of its biomarker potential. Furthermore, being a cross-sectional study, we cannot establish causality.
We have amended the conclusions section (biomarker potential has been removed) and now reads as follows:
(lines: 565-568): In conclusion, our study sought to shed light on the presence of antibodies directed against the NR2 subunit of the NMDA glutamate receptor in MS. Antibody titers were significantly higher in MS patients than in healthy controls, which were associated with disease severity and disability in patients. The activity of NR2 was also correlated with antibody activity levels against coagulant components, which contribute to disease progression, according to our previous studies. In view of these findings, further studies are needed to shed light on anti-NR2 antibodies impact on MS pathophysiology, providing new insights in the perturbation of synaptic mechanisms and signal transmission, as well as ablation of NMDAR function.
Reviewer 3 Report
Comments and Suggestions for Authors
In this study, the authors compared antibodies to the NR2 subunit and antibody activity against coagulant components in the serum of multiple sclerosis (MS) patients and gender-matched healthy controls. The authors concluded that anti-NR2 antibody titers were significantly higher in MS patients than in healthy controls (HCs), associated with disease severity and disability in patients. The activity of NR2 correlated with antibody activity levels against coagulant components, which contribute to disease progression. The idea behind this experiment is quite innovative. But I still have some suggestions about this work.
1. Although the authors verified the higher anti-NR2 in the RRMS group than in HCs with the Mann-Whitney U test, the authors did not compare the clinical profiles of these two groups. After deleting the two groups of patients, SPMS and PPMS, do the age and gender differences between the two groups of patients, RRMS and HCs, exist?
2. As the authors point out, anti-NR2 is associated with many diseases. Therefore, the correlation between anti-NR2 and MS may be less specific. I suggest the authors are more conservative and cautious when concluding that anti-NR2 antibodies may be disease monitoring and prognosis biomarkers.
3. Many of the patients in the study were receiving medication. Whether drug treatment will affect the study results is recommended to be discussed in the discussion paragraph.
Author Response
Reviewer 3
Comment 1. Although the authors verified the higher anti-NR2 in the RRMS group than in HCs with the Mann-Whitney U test, the authors did not compare the clinical profiles of these two groups. After deleting the two groups of patients, SPMS and PPMS, do the age and gender differences between the two groups of patients, RRMS and HCs, exist?
We have included two more columns in Table 1, showing the analysis of clinical features in the RRMS group. Similar with the comparison of all MS with HCs, the individuals in RRMS and HC groups were age (p= 0.53) and gender (p= 0.38) matched.
(Lines 153-160): Table 1 also shows the clinical and non-clinical features of the RRMS course solely since there were analyses conducted specifically in this MS type. RRMS patients and HCs were matched in age (p=0.53) and gender (p=0.38). Fifty-six percent of patients had mild MS disability (EDSS: 0-3.0), 34.7% of MS participants showed moderate disability (EDSS: 3.5-5.5), and 9.3% had severe disability status scale (EDSS: 6.0-9.5). In addition, the dis-ease progression indicated by the MSSS score showed a median index of 3.05 (Inter-quartile Range: 2.1-4.3), with 6.7% of patients stratified into aggressive disease progression (MSSS: 7-10).
Comment 2. As the authors point out, anti-NR2 is associated with many diseases. Therefore, the correlation between anti-NR2 and MS may be less specific. I suggest the authors are more conservative and cautious when concluding that anti-NR2 antibodies may be disease monitoring and prognosis biomarkers.
We have amended the conclusions (biomarker potential removed) and the conclusion section now reads:
(lines: 565-568): In conclusion, our study sought to shed light on the presence of antibodies directed against the NR2 subunit of the NMDA glutamate receptor in MS. Antibody titers were significantly higher in MS patients than in healthy controls, which were associated with disease severity and disability in patients. The activity of NR2 was also correlated with antibody activity levels against coagulant components, which contribute to disease progression, according to our previous studies. In view of these findings, further studies are needed to shed light on anti-NR2 antibodies impact on MS pathophysiology, providing new insights in the perturbation of synaptic mechanisms and signal transmission, as well as ablation of NMDAR function.
Comment 3. Many of the patients in the study were receiving medication. Whether drug treatment will affect the study results is recommended to be discussed in the discussion paragraph.
We have analyzed the antibody levels based on the medication the patients were receiving for MS treatment and we included one subsection in the Results, indicating our analysis. Moreover, we enhanced the results with two additional Tables (Tables 7 and 8) and one Figure (Figure 5).
(Lines: 267-288) 2.4. Analysis between anti-NR2 antibody levels and receiving medications for MS treatment
MS patients receiving medication and those not receiving medication were compared to HCs in terms of binding activity against the NR-2 subunit, considering MS treatment. (Table 7). Using the post-hoc Dunn’s test, there was no significant difference observed between MS patients receiving medication and HCs, MS patients without medication and HCs, or MS patients receiving medication and patients without medication.
However, when analyzing particularly the NR2-reactive antibody levels between patients who received medication and HCs by the Mann-Whitney test, a significantly higher mean activity for NR2 was revealed in patients [mean: 1.78 (SEM: 0.29)] compared to healthy controls [mean: 1.68 (SEM: 0.11); p<0.05].
Upon conducting a more in-depth analysis, it was demonstrated that MS patients taking immunosuppressive medication had higher levels of antibodies (mean: 1.77; SEM: 0.44) compared to healthy controls (mean: 1.68; SEM: 0.11; p<0.05). This was not observed in MS patients taking immunomodulatory medication and compared with controls (Table 8). As Figure 5 shows, NR-2 reactivity levels varied significantly between the patients with immunosuppressive drugs and controls (Figure 5). On the other hand, logistic regression analysis did not reveal an association between antibody presence and medication received in MS group [OR: 0.69 (95% CI: 0.32-1.2)].
Round 2
Reviewer 1 Report
Comments and Suggestions for Authors
I received responses to my remarks. I have no further comments.
Author Response
Thank you for the constructive comments and improvements to the manuscript.
Reviewer 2 Report
Comments and Suggestions for Authors
I think the authors' comments on the questions raised by the reviewers are accurate.
Author Response

(The authors gave the same response as above.)
